# Evolutionary roots of the risk of hip fracture in humans

Hadas Leah Levine[1,2], Nir Shvalb[3], Ariel Pokhojaev[1,2,4], Samuel Francis[1,2], Ruth Pelleg-Kallevag[1,2,5], Victoria Roul[1,2], Jean-Jacques Hublin[6,7], Frank Rühli[8] & Hila May [1,2✉]

The transition to bipedal locomotion was a fundamental milestone in human evolution. Consequently, the human skeleton underwent substantial morphological adaptations. These adaptations are responsible for many of today's common physical impairments, including hip fractures. This study aims to reveal the morphological changes in the proximal femur, which increase the risk of intracapsular hip fractures in present-day populations. Our sample includes chimpanzees, early hominins, early *Homo* Neanderthals, as well as prehistoric and recent humans. Using Geometric Morphometric methods, we demonstrate differences in the proximal femur shape between hominids and populations that practiced different lifestyles. We show that the proximal femur morphology is a risk factor for intracapsular hip fracture independent of osteoporosis. Changes in the proximal femur, such as the shortening of the femoral neck and an increased anterolateral expansion of the greater trochanter, are associated with an increased risk for intracapsular hip fractures. We conclude that intracapsular hip fractures are a trade-off for efficient bipedal walking in humans, and their risk is exacerbated by reduced physical activity.

[1] Department of Anatomy and Anthropology, Sackler Faculty of Medicine, Tel Aviv University, Tel Aviv 6997801, Israel. [2] The Shmunis Family Anthropology Institute, the Dan David Center for Human Evolution and Biohistory Research, Sackler Faculty of Medicine, Tel Aviv University, Tel Aviv 6997801, Israel. [3] Mechanical Engineering Department, Ariel University, Ariel 40700, Israel. [4] Department of Oral Biology, The Maurice and Gabriela Goldschleger School of Dental Medicine, Tel Aviv University, Tel Aviv 6997801, Israel. [5] Zefat Academic College, Zefat, Israel. [6] Chaire de Paléoanthropologie, CIRB (UMR 7241 – U1050), Collège de France, Paris 75231, France. [7] Max-Planck Institute for Evolutionary Anthropology, Leipzig 04103, Germany. [8] Institute of Evolutionary Medicine, University of Zurich, Zurich CH-8057, Switzerland. ✉email: mayhila@tauex.tau.ac.il

The transition to bipedal locomotion was one of the earliest, most significant, and fundamental adaptations of the hominin lineage. Since this transition was a nonlinear process, it raised considerable debate regarding its time of appearance and causes[1–3]. Nonetheless, walking on two legs required a considerable anatomical rearrangement of the entire body, specifically that of the proximal femur (PF), to enable the balancing of the trunk over a single supporting limb during locomotion with minimal energy consumption[4–7]. However, different hominins displayed a varying combination of traits, which may indicate a mixed locomotion pattern or exclusive bipedality (e.g., refs. [8,9],). Generally, changes in PF morphology during human evolution included an increase in the size of the femoral head, a shortening and an increase in the height of the femoral neck, and narrowing of its antero-posterior dimension, an increase in the lateral projection of the greater trochanter, a more medially oriented lesser trochanter, and an increase in the neck shaft angle (NSA)[8,10–20]. These morphological differences were captured using different methodologies, mainly linear measurements. Although multivariate analyses of linear measurements obtained from the PF could distinguish between different groups of hominids[8,13,20], landmark-based geometric morphometric (GM) analysis yielded better results[20,21]. Researchers also suggested that PF morphology continued to change among humans during the Holocene (approximately in the last 10,000 years) due to lifestyle changes, since a positive correlation was found between NSA and increased sedentism[22,23].

Although the transition to bipedal locomotion most likely resulted from positive selective pressure[24–29], the morphological adaptation that was required to withstand the forces applied to the hip might involve some compromises[6,30], namely, an increased risk for hip fracture[6,31–34]. However, contradictory results exist regarding the association between the risk of sustaining a hip fracture and PF measures, such as hip/neck axis length[35–42], NSA[35,38,40–42], and femoral neck width[35,36,42]. Neverthless, Gregory and Aspden[32] suggested that combining these measures or capturing the entire shape of the PF may shed more light on this association. Furthermore, two-dimensional shape analyses of the PF revealed a high discrimination power between individuals with and without a hip fracture[43].

Hip fracture is a pathology exclusive to humans, with increased incidence over time[44]. The two most common types are intracapsular (ICHF; at the femoral neck) and extracapsular (at the intertrochanteric region) hip fractures[45]. Owing to its high prevalence worldwide, hip fracture is considered a major public health concern, as well as an economic and social burden[46,47].

Although the pathogenesis of hip fracture is multifactorial, it can be divided into two major groups: fractures associated with low bone mass density and those associated with an increased risk of falling. Falling is thought to have a more significant effect on the risk of fracturing the hip than is osteoporosis[45,47]. Hence, PF shape might play an important role in determining the risk of fracturing the hip[32,43]. Accordingly, recognizing the fundamental risk factors, such as bone morphology, is of major importance for improving our ability to predict the risk of fracture and to develop new, effective preventive measures.

The major aims of the current study were as follows: (1) To determine whether the morphological changes in the PF, which were initiated with the transition to bipedal locomotion, continued in modern humans following changes in their lifestyle. (2) To determine whether the morphological adaptation of the PF is associated with an increased risk for hip fractures in recent humans. Indeed, changes in the PF morphology, such as the shortening of the femoral neck and an increased anterolateral expansion of the greater trochanter, identified between early hominins and prehistoric humans and Neanderthals, have continued among modern humans. These changes are more pronounced in individuals with non-osteoporotic ICHF. Accordingly, hip morphology can be related as a risk factor for ICHF, independent of osteoporosis.

## Results

**Reliability analyses.** Intra- and inter-observer errors in landmark placement of the GM protocol developed for this study (see the method section) were negligible (Fig. S1).

**Shape variance of the PF between hominids.** The PF shape was found to be sex independent for both humans and *Pan troglodytes*. In the recent human (living) sample, sex explained only 2.2% of PF shape variation and did not differ significantly between males and females (Table 1 and Fig. 1). In *Pan troglodytes*, sex explained 4.5% of PF shape variance with no significant differences between the sexes (Fig. S2). Although we could not test for sexual dimorphism in PF shape among the early hominins, the results obtained from both humans and *Pan troglodytes* supported our decision to combine males and females in further analyses. Since the life expectancy of the fossils and the archaic sample was lower than in recent humans (Table S1), we limited the age range of the recent human sample to be between 18 and 45 years (Table S2) when comparing prehistoric and living groups. Procrustes ANOVA indicated that group membership is

**Table 1 Procrustes ANOVA analyses for proximal femoral.**

| | Df | SS | MS | R² | F | Z | p |
|---|---|---|---|---|---|---|---|
| Human Evolution: Shape variance among hominin groups | | | | | | | |
| log(Csize) | 1 | 0.1508 | 0.1508 | 0.1305 | 21.591 | 5.930 | 0.001 |
| Group | 4 | 0.3131 | 0.0783 | 0.2709 | 11.204 | 6.030 | 0.001 |
| log(Csize):Group | 4 | 0.0560 | 0.0140 | 0.0485 | 2.005 | 2.493 | 0.004 |
| Terminal Pleistocene-Holocene Levant: Shape variance for different subsistence strategies | | | | | | | |
| log(Csize) | 1 | 0.027468 | 0.027468 | 0.037449 | 4.129171 | 3.953799 | 0.002 |
| Group | 3 | 0.072466 | 0.024155 | 0.098797 | 3.631166 | 5.535288 | 0.001 |
| log(Csize):Group | 3 | 0.014894 | 0.004965 | 0.020306 | 0.746321 | −1.14585 | 0.884 |
| Recent humans: Shape variance among recent humans without ICHF | | | | | | | |
| log(Csize) | 1 | 0.028086 | 0.028086 | 0.020563 | 4.496895 | 4.311102 | 0.001 |
| Sex | 1 | 0.030804 | 0.030804 | 0.022553 | 4.932006 | 4.755846 | 0.001 |
| Age | 1 | 0.040903 | 0.040903 | 0.029947 | 6.548896 | 4.598008 | 0.001 |
| log(Csize):Sex | 1 | 0.005069 | 0.005069 | 0.003712 | 0.811662 | −0.49147 | 0.695 |
| log(Csize):Age | 1 | 0.005956 | 0.005956 | 0.00436 | 0.953539 | 0.093149 | 0.450 |
| log(Csize):Sex:Age | 1 | 0.007164 | 0.007164 | 0.005245 | 1.147082 | 0.580312 | 0.282 |

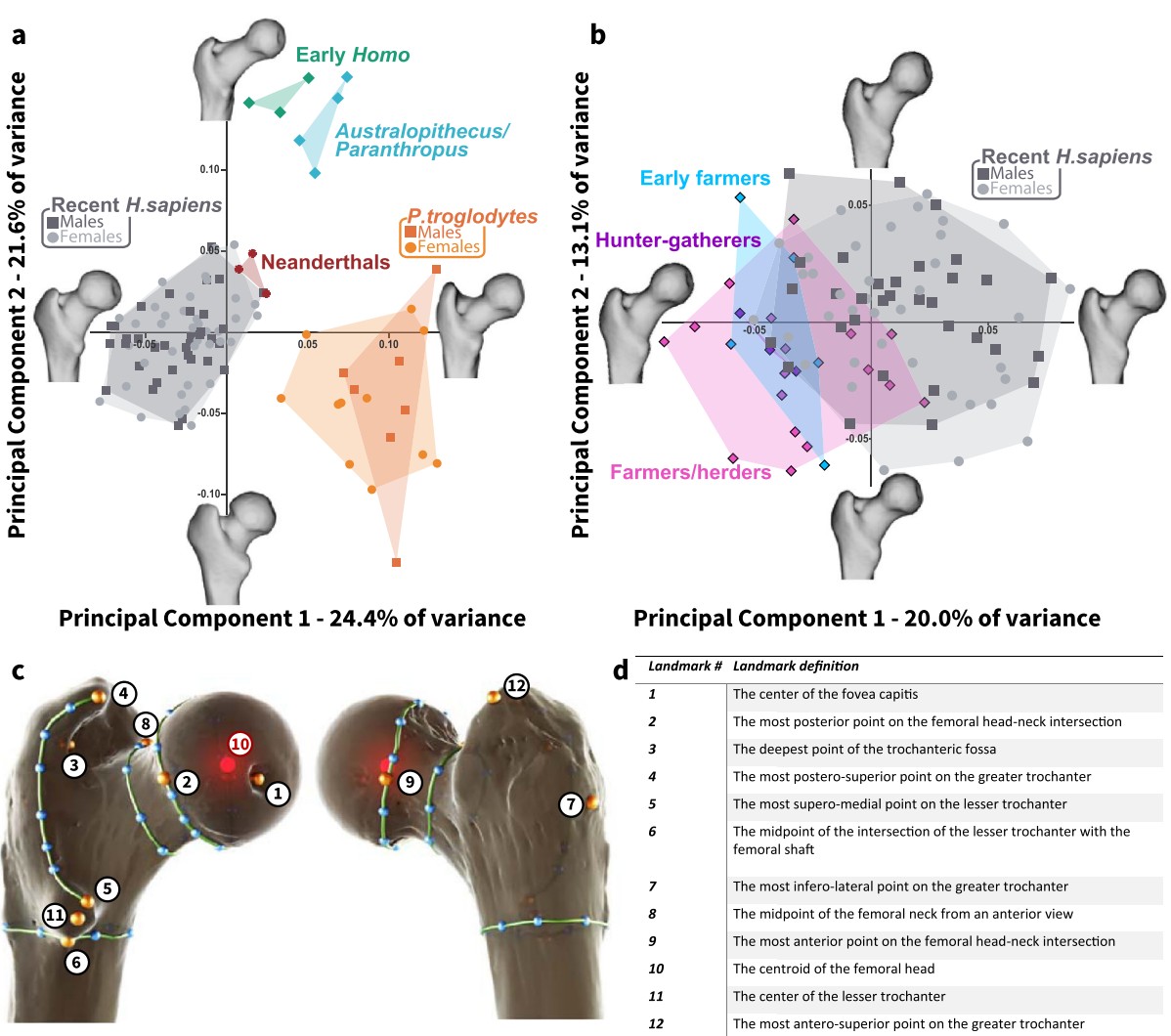

**Fig. 1 Proximal femoral shape variance among Pan troglodytes, early hominin, and modern human groups. a** Principal component analysis (PCA) plot in shape-space for the proximal femur of *Pan troglodytes* (orange), *Australopithecus/Paranthropus* (light blue), early *Homo* (light green), Neanderthals (red), and recent humans (gray; age 18–45 years). PC1 (explains 24.4% of the shape variance) distinguishes between *Pan troglodytes* ($N = 18$) and recent humans ($N = 74$); early hominins ($N = 4$) and early *Homo* ($N = 3$) fell in between and can be distinguished from other groups along PC2 (explains 21.6% of the shape variance). Neanderthals are in the upper variation of recent humans along both PC1 and PC2. **b** PCA in shape-space for the proximal femur among populations with different subsistence strategies: Hunter-gatherers ($N = 3$) – Epi-Paleolithic (purple), early farmers ($N = 5$) – Pre-Pottery Neolithic (blue), pastoralists ($N = 17$) – Chalcolithic (pink), and recent humans (gray; 18–45 years old, $N = 70$). A gradient of change in the proximal femoral shape variance over time is evident along PC1. Details regarding the sample included in the study appear in Tables S1 and S2. **c** The positions of landmarks (orange, numbered dots), curves (green lines), and semi-landmarks (blue dots) on the proximal femora. **d** Definition of the landmarks' position (definitions of the curves' position are presented in Table 2).

the main predictor of PF three-dimensional (3D) shape variation (it explained 27.1% of the shape variance; Table 1).

The variation in the PF 3D shape among *Pan troglodytes*, early hominins, early *Homo*, Neanderthals, and recent humans is presented in Fig. 1a. The first principal component (PC1), which explained 24.4% of the PF shape variance, significantly distinguished between *Pan troglodytes* and recent humans (a pairwise comparison with a false discovery rate (FDR) correction: $p$-adj. = 0.025; Table S3). However, PF size differences did not dictate the differences in PF shape between them (Fig. S3). Early hominins (*Australopithecus/Paranthropus*) and early *Homo* fell in between these groups along PC1. However, they could be distinguished from them along PC2, which explained 21.6% of the PF shape variance (Fig. 1a). Neanderthals fell at the upper limit of recent human variation, between recent humans and *Pan troglodytes*, along PC1 and between recent humans and early

hominins and early *Homo* along PC2 (Fig. 1a). Furthermore, their PF shape variation fell within that of the prehistoric populations of the Levant, at the edge of the early farmers' group and overlapped that of the hunter-gatherers (Fig. S4).

Along PC1, from *Pan troglodytes* (positive values) to recent humans (negative values), the femoral head became rounder and retroverted, the femoral neck became shorter and wider (antero-posteriorly), and the trochanteric fossa became shallower. The greater trochanter became lower relative to the femoral head, decreased its vertical dimension, and rotated postero-medially while increasing its lateral flaring. The intertrochanteric crest expanded medially towards the posterior aspect of the femoral head. The lesser trochanter became less prominent and more medially positioned (Fig. 1a). Along PC2, from early hominins and early *Homo* (positive values) to recent humans (negative values), PF shape changes included a femoral head that became

rounder, larger, and superiorly projected, along with the shortening of the femoral neck with a larger concavity at its superior aspect, an increase in lateral flaring of the greater trochanter, and the lesser trochanter became more prominent (Fig. 1a). It is noteworthy that these changes along the PC axes do not represent a linear development throughout human evolution but rather, differences between groups.

**Shape variance of the PF among humans with different lifestyles.** The 3D shape variation of the PF among human groups, characterized by different subsistence strategies (i.e., hunter-gatherers, early farmers, pastoralists, and recent humans), is presented in Fig. 1b. Along PC1, which explained 20.0% of the shape variance, the hunter-gatherers and early farmers overlapped and fell at the margins of the PF shape variation of recent humans. The pastoralists exhibited a larger shape variation than that of the prehistoric populations (hunter-gatherers and early farmers), and it partially overlapped with the variation of recent humans (Fig. 1b). Shape changes along this axis, from prehistoric/protohistoric to recent times, are expressed in a slight retroversion of the femoral head, shortening of the femoral neck, increased anterolateral expansion of the greater trochanter, postero-medial expansion of the intertrochanteric crest, and a slightly more medially positioned lesser trochanter (Fig. 1b).

According to the Procrustes ANOVA results, group membership explained 9.8% of the shape variation, whereas size explained only 3.7%. Size, however, did not account for the allometric shape differences among humans with different subsistence strategies (Table 1).

**Proximal femoral shape variation among recent humans with and without a hip fracture.** To examine the effect of the demographic characteristics (sex and age) as well as the PF size on the PF shape, we carried out Procrustes ANOVA for the entire control population (i.e., recent humans without an ICHF; $N = 206$). Centroid size (Csize), sex, age, and the interaction between sex and age explained between 0.8 and 3.0% of the shape variation (see R2, Table 1) with no significant differences between males and females regarding the PF shape (pairwise comparison: $p = 0.767$). To examine how osteoporosis affected the shape variation among recent humans with and without an ICHF, we carried out analyses on a subsample with DEXA scores ($N = 59$). This subsample was divided into groups according to the health status of their bone (healthy, osteopenic, or osteoporotic; see the "Materials" section and Table S2) and the manifestation of ICHF (i.e., yes/no).

A between-group PCA (bgPCA) indicated that the non-osteoporotic fracture group could be distinguished from the other groups along PC1 (Fig. 2a). To determine whether differences in shape variation were real and did not result from a "high p/n" setting[48,49], we calculated the average Procrustes distances of each individual by a group, from the mean shape of each group and presented these distances in a heatmap and a dendrogram (Fig. 2b). Individuals with osteopenia or osteoporosis were clustered together, regardless of the fracture. Individuals with a 'normal' hip were closer to the osteopenic groups. The non-osteoporotic ICHF group was the most distant and was clustered in a different branch, separated from the other groups.

The PF shape differences between these groups were visualized by superimposing the mean shape of each pathologic group on the mean shape of the control group (no osteopenia/osteoporosis and no ICHF). Accordingly, the largest shape differences were for the mean shape of the non-osteoporotic ICHF group, and the smallest shape differences were for the osteopenic group (Fig. 2c and Fig. S5). The osteoporotic mean shape yielded larger

differences, especially for the non-fractured group, compared with those obtained for the osteopenic mean shapes (Fig. 2c). The mean shape of the non-osteoporotic ICHF group, when compared to the mean shape of the control group, was characterized by shortening of the femoral neck and a slight increase in its height. The greater trochanter was expanded antero-laterally, and its most lateral point was positioned more antero-superiorly. The trochanteric fossa was shallower, and the lesser trochanter was slightly more prominent infero-medially. In contrast, other morphological changes were identified in the non-fractured osteoporotic group. These changes were expressed mainly in the shape of the femoral neck, which became narrower antero-posteriorly at its proximal end (Fig. 2c and Fig. S5).

**Human evolution, lifestyle, and risk of ICHF.** To determine how differences in PF morphology between hominins, as well as between human groups that practiced different lifestyles, are associated with non-osteoporotic ICHF risk, we examined PF shape variance among early hominins, early *Homo*, Neanderthals, prehistoric/protohistoric groups (Epi-Paleolithic hunter-gatherers, PPN early farmers, and Chalcolithic pastoralists) (Table S1), recent humans with a non-osteoporotic ICHF, as well as recent humans from the control group (non-osteoporotic, not fractured) (Table S2). Considering that age explains some of the variations in PF shape (Table 1) and that the differences in life expectancy between recent and ancient humans are large, we included in the analyses relatively young individuals with ICHF (50 to 60 years old) and two age groups from the control sample: individuals 20 to 40 years old (corresponding to ancient groups), and individuals 50 to 60 years old (corresponding to the non-osteoporotic ICHF group) (Table S2).

The PF shape of Neanderthals, prehistoric/protohistoric humans, and the non-osteoporotic ICHF group fell within the variation of recent humans in the first two PCs, which together explained 41.2% of the shape variance (Fig. 3a). However, the Neanderthal and prehistoric/protohistoric proximal femora could be differentiated from the non-ICHF group along PC1 (Fig. 3a). The average Procrustes distances of the early hominins and early *Homo* were the most distant from the mean shape of the non-osteoporotic ICHF group and the least distant from the Neanderthals and prehistoric/protohistoric groups (Fig. 3b). According to the dendrogram, the various groups were organized in two major clusters: the first consisted of early hominins and early *Homo*, and the second consisted of Neanderthals and humans. Neanderthals were more distant from the non-osteoporotic ICHF group than was the prehistoric/protohistoric group. However, both were clustered together and were more distant from the non-osteoporotic ICHF group, compared with recent humans (Fig. 3b). Furthermore, none of the prehistoric femora were classified in the ICHF group, although 11.3% were classified as recent humans. However, 15.5% of the recent femora were classified in the non-osteoporotic ICHF group (Table S4).

The shape differences between these groups were visualized by superimposing the mean shape of the control sample (no fracture) and the non-osteoporotic ICHF sample on that of the prehistoric/protohistoric sample. In general, the mean shape of recent individuals (with and without ICHF) showed similar trends in PF shape differences; however, the magnitude of the differences was larger among the non-osteoporotic ICHF group (Fig. 4). Modifications in shape mainly consisted of retroversion of the femoral head, expansion of the greater trochanter antero-laterally, with its most prominent lateral point shifting antero-superiorly, and the intertrochanteric ridge expanded medially. In addition, the trochanteric fossa was shallower and the lesser trochanter was positioned more infero-medially. The combination of femoral head rotation,

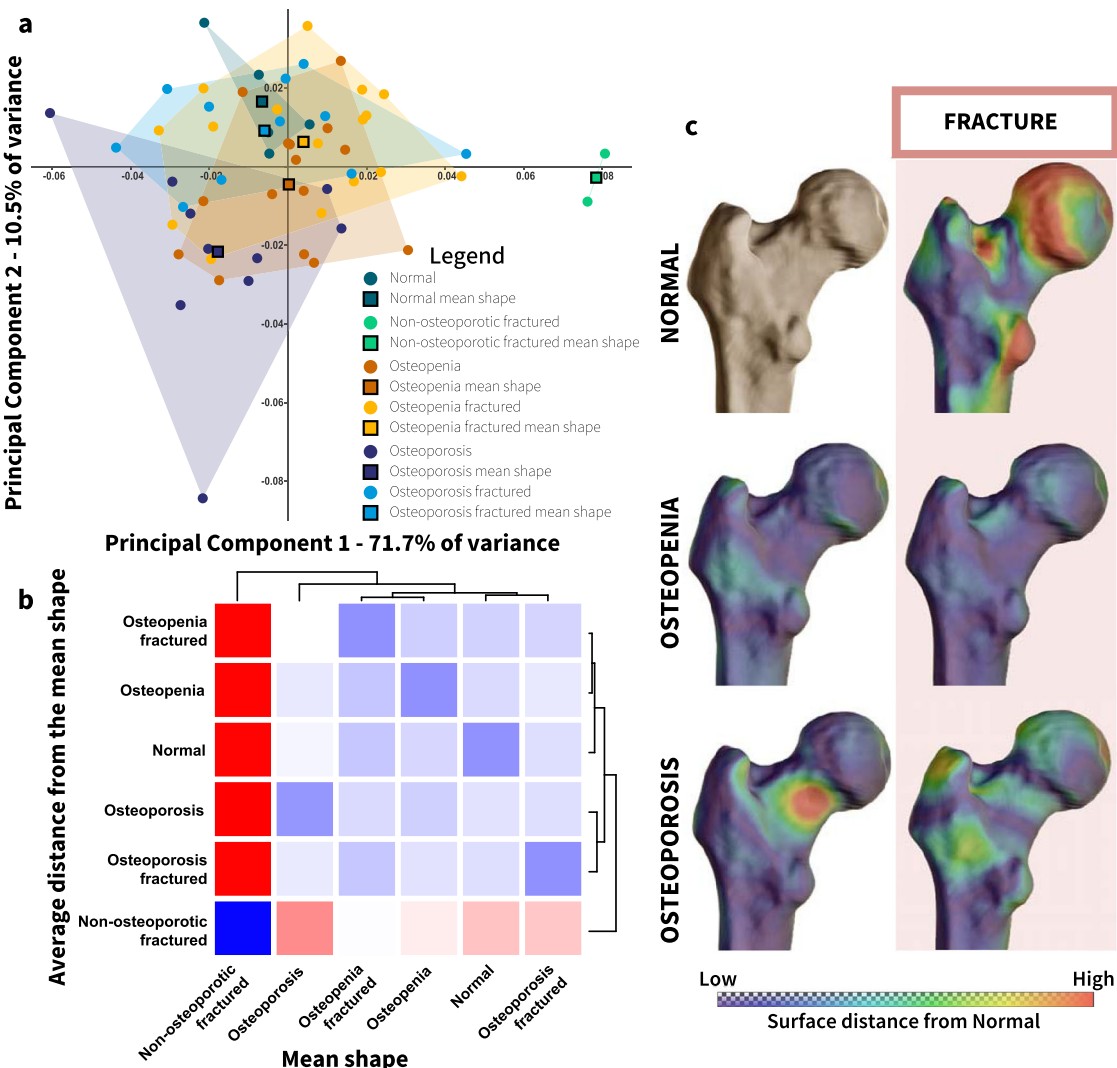

**Fig. 2 Proximal femoral shape variance among living humans by osteoporosis and intracapsular hip fracture. a** Between-group PCA of recent humans with and without an intracapsular hip fracture (ICHF), by osteoporosis diagnosis obtained from the DEXA scores ($N = 59$). The non-osteoporotic ICHF group can be distinguished from the other groups along PC1. **b** Heatmap and dendrogram created from the average Procrustes distances of individuals belonging to the same group from the mean shape of each group. **c** The surface distance of the mean shape of each group from the mean shape of the control group (non-osteoporotic, not fractured) represented by colors (blue for the lowest differences and red for the largest differences).

widening of the femoral neck anteriorly, and the medially positioned intertrochanteric ridge resulted in an increased concavity of the femoral neck at its posterior aspect.

**Discussion**

This study is the first to examine whether differences in PF shape between various hominins, as well as between human groups characterized by different lifestyles, are associated with a higher risk of ICHF in recent humans, independent of osteoporosis. Furthermore, we have demonstrated that the 3D shape of the PF, as opposed to its size[50], is independent of sex for both humans and *Pan troglodytes*. Sexual dimorphism related to PF size is also assumed for early hominins[51], although no data exist regarding its shape. However, our results strengthened our assumption that sexual dimorphism is a minor factor in dictating the PF shape in early hominins since both groups (early hominins and early *Homo*) grouped together in the PCA and could be distinguished from other groups. These results highlight our ability to compare different groups without controlling for sex.

Our study is in agreement with previous studies regarding the morphological traits that differentiate *Pan troglodytes*, early

hominins, early *Homo*, and recent humans (e.g., refs. [13,20,21]). Changes such as shortening of the femoral neck, an increased NSA, and a larger femoral head were among the characteristics that differentiated early hominins and early *Homo* from recent humans (e.g., refs. [8,11,13,14,16,17,20,52]). A larger femoral head, a taller neck, a shallower trochanteric fossa, and a greater trochanter with lower height and larger lateral expansion were among the traits that differentiated early hominins, early *Homo*, and recent humans from *Pan troglodytes* (e.g., refs. [8,20,21,52]). Furthermore, as previously suggested[17], we also did not find a gradual linear transformation in PF morphology from *Australopithecus* or an *Australopithecus*-like ancestor of *Homo sapiens*. However, there are large gaps in the availability of the PF of fossils between these groups in order to verify this notion. The compatibility between our results and previous ones strengthens the applicability of our landmark-based GM protocol for testing hypotheses related to differences in PF shape between human groups experiencing different lifestyles and between humans and Neanderthals.

Explanations regarding the differences in PF morphology between *Pan troglodytes* and the genus *Homo*, as well as between

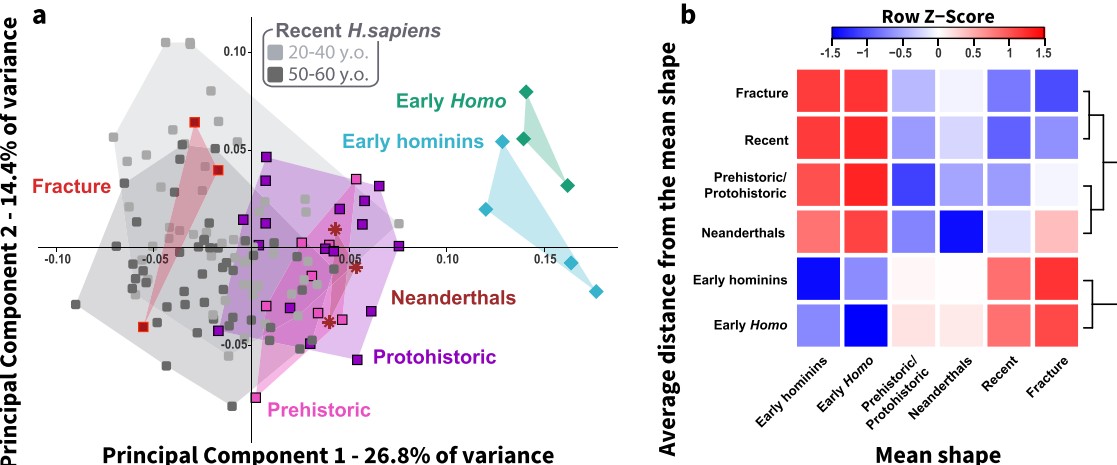

**Fig. 3 Proximal femoral shape variance among early hominins, Neanderthals, and human groups (with and without hip fracture). a** Principal component analysis (PCA) plot in shape-space for the proximal femur of early hominins (light blue; $N = 4$), early *Homo* (light green; $N = 3$), Neanderthals (brown; $N = 3$), prehistoric (hunter-gatherers and early farmers, pink; $N = 8$), protohistoric (pastoralists, purple; $N = 17$), recent humans with non-osteoporotic fractures (red, $N = 3$), and recent humans (control group; aged 20–40 and 50–60 years old, gray; $N = 101$). PC1 (explains 26.8% of variance) distinguishes early hominins and early *Homo* from Neanderthals and modern humans. The non-osteoporotic fracture group fell on one side of the recent human cloud, whereas the Neanderthals, Prehistoric, and Protohistoric proximal femora were on the other side. **b** Heatmap and dendrograms produced based on the average Procrustes distances of each group from the mean shape of each group. The Prehistoric/Protohistoric group included Epi-Paleolithic hunter-gatherers, PPN early farmers, and Chalcolithic pastoralists. Recent humans included individuals aged 20–40 and 50–60 years. The fracture group included individuals with non-osteoporotic ICHF.

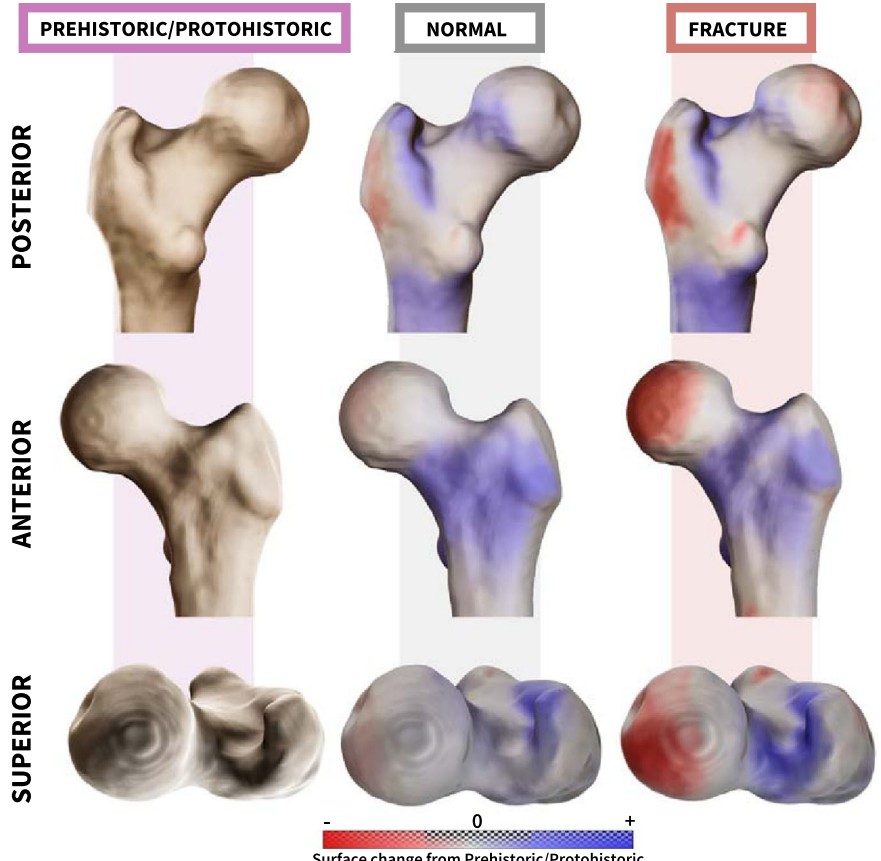

**Fig. 4 Superimposition of the proximal femoral mean shape of recent humans (with or without a non-osteoporotic fracture) on the proximal femoral mean shape of the Prehistoric/Protohistoric group.** The superimposition of the mean shapes from posterior, anterior, and superior views. The colors indicate the magnitude of shape change (i.e., the distance between surfaces) between the mean shape of the Prehistoric/Protohistoric surface (white) and that of the superimposed one (recent with or without a fracture).

hominins, vary. Most researchers agree that these morphological differences are associated with differences in the biomechanical forces applied to the hip due to different posture and locomotion patterns[3,20,53]. However, it was also argued that no direct relationship exists between stride characteristics (e.g., pelvic rotation) and anatomical morphology[54]. Furthermore, contradictory opinions exist regarding the reasons for differences in PF shape between early hominins and recent humans. Some researchers attribute them to different modes of walking or a combination of modes of locomotion (refs. [9,14,31,55–60]). Others, however, claim that the anatomical organization of early hominins' pelvis and lower limbs allowed them to have a mode of walking similar to ours, with no differences in energetic efficiency or compromises due to gaining new characteristics (e.g., refs. [52,54,61–65],). According to Lovejoy[30], the morphological changes in the PF from early hominins to modern humans resulted from an increase in brain size during human evolution, which required a rearrangement of the pelvis to enable childbirth without reducing the efficiency of bipedal locomotion. Nevertheless, ref. [66] demonstrated that the morphological adaptations that were required for bipedal locomotion (even in uncommitted bipeds) were associated with a reduced ratio of brain-to-body size as early as in the *Australopithecus*.

Our study sheds new light on Neanderthal PF shape variation. We have demonstrated that it fell at the margins of recent human variation and overlapped with the PF shape variation of the Epi-Paleolithic hunter-gatherers. Based on the bone functional adaptation concept, bones react to the loadings applied to them to sustain the stress, meaning that the plasticity of bones in response to strain results in an adaptation of the bone shape[67,68]. Hence, it is commonly used for comparing past population behavior[69,70]. Consequently, the similarity in the PF shape variation of Neanderthals and Epi-Paleolithic hunter-gatherers suggests that a similar pattern of loadings existed on their bones; this supports previous findings indicating that Neanderthals were well adapted to hunting[65,71]. Furthermore, differences in PF shape variation among humans (e.g., hunter-gatherers, early farmers, pastoralists, and recent humans) might reflect different patterns and magnitudes of loadings applied to their lower limbs following the transition to a more sedentary way of life[69,72]. This is further supported by previous studies that demonstrated that NSA increased with increased sedentism[22]. However, considering the sensitivity of the 3D shape analysis of the PF for identifying subtle morphological differences[20,21], we found that increased sedentism mainly involved changes in the femoral neck (shortening and widening antero-posteriorly) and greater trochanter (e.g., an increased anterolateral projection) rather than the orientation of the neck and head in relation to the shaft.

Although the association between the morphology of the PF (estimated by femoral neck length and width, and NSA) and the ICHF risk in recent humans has been vastly debated[35–42], we have demonstrated that recent humans are at a higher risk for ICHF than prehistoric humans and Neanderthals. Furthermore, we could identify the morphological features that increase the risk of ICHF in recent humans. Specifically, the femoral neck was more concave, and the greater trochanter was expanded more antero-laterally.

The most common mechanism of hip fracture in humans is a sideways fall, which directly impacts the greater trochanter and places the femoral neck at a high risk for fracture. Chimpanzees, however, will rarely fracture their hip following a fall[73]. Previous studies associated this increased risk in humans with the thinning of the supero-lateral aspect of the femoral neck, which occurs with advanced age; thus, the femoral neck cannot withstand the improper stress applied to it due to the fall[74–76]. Our results indicated that the PF shape also contributes to the risk of

fracturing the hip in a sideways fall. Theoretically, the PF's optimal, most resistant geometry that could withstand the larger compressive and tensile forces on the femoral neck in a sideways fall would be an I-beam architecture[77,78]. In this model, the lateral apex and the center of the femoral head are aligned on the same plane, and the beam (i.e., the femoral neck) connecting them is straight, resembling the PF morphology of early hominins and early *Homo*. Obviously, the human PF shape was not adapted to withstand lateral impact but rather, to better withstand forces related to bipedal locomotion (e.g., the superior position of the femoral head relative to the greater trochanter) and, as such, decreasing the risk for a femoral neck stress fracture[79,80]. Accordingly, the larger the deviation from the I-beam architecture, the greater the risk of manifesting an ICHF following a sideways fall, regardless of osteoporosis. The risk of fracture increases since the PF is less resistant to the reverse of forces applied to the femoral neck due to the fall[77]. In our study, the non-osteoporotic ICHF group had a morphology that deviated the most from the I-beam architecture (early hominin/*Homo*).

According to our results, the major morphological modifications in the PF between groups representing different phases in human evolution, and by adopting a modern way of life (less physically active) were concentrated in the muscle attachment areas. These modifications include the insertion sites of the abductor muscles on the antero- and postero-lateral aspects of the greater trochanter; the lateral rotators and/or abductors of the thigh when it is flexed on the superior, posterior, and medial aspects of the greater trochanter; and the major flexor of the hip, which attaches to the lesser trochanter[81]. In recent populations, who generally walk and run less, reduced forces are applied to the PF. Consequently, the greater trochanter flaring shifted slightly anteriorly, whereas its superior and posterior aspects rotated medially, and the femoral neck became more concave. The trochanteric fossa was flattened, and the lesser trochanter became more medially positioned. Although these changes are more suitable to withstand the forces applied on the femur in a bipedal stance and locomotion[75,82] and increase the efficiency of bipedal locomotion by enabling greater lateral stabilization afforded by the abductor muscles[83], they also increase the risk for ICHF in a sideways fall.

The non-osteoporotic ICHF group fell within the variation of recent humans without a fracture (although it was situated relatively at its margin) and could be distinguished from the prehistoric group. Accordingly, among modern humans, the risk of ICHF is increased. Notwithstanding, most individuals, especially young ones, do not fracture their hip following a sideways fall to the ground, even when their morphology is within the variation of the non-osteoporotic ICHF group. This is probably due to the microscopic characteristics of their bones, which can withstand the impact[75]. However, the combination of advanced age, leading to deterioration in the bones' microscopic characteristics[84], and the above morphology most likely increases the risk for ICHF. It is noteworthy that the osteoporotic PF exhibited other morphological differences (mainly narrowing of the proximal part of the femoral neck at its antero-posterior aspect), regardless of the existence of a fracture. This characteristic suggests another risk factor for hip fracture in the osteoporotic group, such as a fracture during standing loadings[85,86]. To summarize, this study suggests that ICHF is a trade-off for efficient bipedal walking in modern humans and that the risk of hip fracture is exacerbated by reduced physical activity and a sedentary lifestyle.

However, this study experienced some limitations. Among the common limitations of paleoanthropological studies is the sample size of fossils. This factor also limited our research and prevented us from drawing conclusions regarding continuous changes in PF shape during human evolution. Furthermore, an additional

| Table 2 Definitions of curve positions. | |
| --- | --- |
| **Curve name** | **Curve definition** |
| 1- Femoral head | A manual curve placed on the femoral head-neck intersection, between landmarks 2 and 9 (Fig. 1c, d). |
| 2- Intertrochanteric crest | A manual curve placed on the intertrochanteric crest, starting at landmark 12 (Fig. 1c, d). |
| 3- Femoral neck | A semi-automatic curve placed around the femoral neck at its midpoint. An anchor point is placed at landmark 8, and then two additional points are chosen on the anterior and posterior aspects of the femoral neck at its midpoint. The program calculates the area of all curves that pass through the anchor point and that are ±15° from the two selected points, and then chooses the curve with the smallest cross-sectional area. |
| 4 - Subtrochanteric | A semi-automatic curve placed around the femoral shaft below the lesser trochanter. An anchor point is placed at landmark 6, and then two additional landmarks are placed on the medial and lateral aspects of the femoral shaft at the same level as the anchor point. The program calculates the area of all curves that pass through the anchor point and that are ±15° from the two selected points, and then chooses the curve with the smallest cross-sectional area. |

Figure 1 presents details about the landmarks' position and its definition.

limitation mainly concerning the early hominin samples is the application of GM protocols using many landmarks on small sample sizes, thus restricting the statistical power and the related conclusions. Future studies should be carried out to test the model suggested here regarding the increased ICHF risk in modern humans.

## Methods

**Materials**. This study included virtual reconstructions of the femora of *Pan troglodytes*, African Lower Paleolithic hominins, European Neanderthals, and humans (prehistoric and recent). CT scans of 18 *Pan troglodytes* were obtained from the Digital Morphology Museum, KUPRI (http://dmm.pri.kyoto-u.ac.jp/dmm/WebGallery/index.html). CT scans of original fossils including an *Australopithecus afarensis* (A.L. 288-1), two *Paranthropus robustus* (SK82 and SK97), an unspecified taxon related to either *Australopithecus afarensis* or *Paranthropus robustus* (KNM-ER 1503), three early *Homo* (KNM-ER 1472, KNM-ER 1481, and KNM-WT15000), and three Neanderthals (Krapina 213, Krapina 214, and Neanderthal 1) were obtained from the digital collection of the Max Planck Institute for Evolutionary Anthropology, Leipzig, Germany. The study of the fossils' CT scan was in accordance with relevant permits (see the "Acknowledgment" section). CT scans (Brilliance 64, Philips Medical System, Cleveland, Ohio) or surface scans (Solutionix-Rexcan SC + 2) of 25 prehistoric and protohistoric adult humans (completely fused femoral head) from the Anthropological Collection, Dan David Center for Human Evolution and Bio-history Research, Sackler Faculty of Medicine, Tel Aviv University, were included as well (Table S1). These Levantine proximal femora were divided into three groups, which were characterized by different subsistence strategies: Prehistoric Epi-Paleolithic hunter-gatherers (ca. 19,000-11,000 cal BP; $N = 3$), Prehistoric Final Pre-Pottery Neolithic early farmers (9250–8000 cal BP PPNC; $N = 5$), and Protohistoric Chalcolithic pastoralists (6000-5300 BP; $N = 17$).

The proximal femora of 307 recent humans (188 females and 119 males aged 18–85 years) were obtained from medical CT scans. Of these scans, 82 individuals, 61 females (74.0 ± 6.56 years old), and 21 males (71.0 ± 8.37 years old), manifested an ICHF (the study group).

For the control group (no ICHF), the surface of the left femur was reconstructed. For the study group, the non-fractured femur was included in the study (either left or right), assuming that the right and left proximal femora are symmetrical[87]. Therefore, for those individuals from the study group whose right side was analyzed, it was mirrored following segmentation via Amira software (v. 6.3; www.fei.com).

When available, DEXA T-scores were obtained from the patient's medical file (i.e., 59 individuals—29 with a hip fracture and 30 without). Then, these individuals were divided into three groups based on their bone health category: healthy = >−1, −1.0<osteopenic < −2.5, and osteoporotic ≤ −2.5[88]. For the rest, information about bone health was obtained from their medical file. Furthermore, we assumed that individuals younger than 60 years did not suffer from osteoporosis[89] unless documented otherwise. Consequently, the study included five individuals with non-osteoporotic/osteopenic ICHF (three individuals between 50 and 60 years old, and two aged 75 and 78 years old).

CT scans of recent humans were carried out before the study (between 2010 and 2017) at the Carmel Medical Center, Haifa, Israel (Brilliance 64, Philips Medical System, Cleveland, Ohio) for medical purposes unrelated to the study. The study (studying femoral morphology from anonymized CT scans and obtaining data from the patient's medical file) was approved by the ethics committee of Carmel Medical Center (approval #432985) as well as by the ethics committee of TAU (approval #0000252-1). All patients have signed informed consent before conducting the scan. The ethical committees required no additional informed consent for this study. The exclusion criteria of this study were as follows: incompletely fused femoral head, bone pathologies (e.g., bone neoplasms and hip

joint arthroplasty not related to hip fracture), locomotor disability acquired before the fracture, or the individual had undergone amputations to the lower limb. In addition, only fossils and prehistoric samples having all landmarks were included in the study.

**Surface generation**. The surface of the PF was generated either by segmentation from the CT stacks using Amira (v. 6.3; www.fei.com) or by surface scanning (Solutionix-Rexcan SC + 2; Solutionix, Seoul, South Korea). For the latter, the reconstruction and alignment of the 3D surface of the bone were carried out using EZScan 7 software (Solutionix, Seoul, South Korea).

**The three-dimensional landmark-based geometric morphometric protocol**. The 3D shape of the PF was represented using 12 landmarks and 31 curve semi-landmarks (which were placed on four curves) (Fig. 1c, d and Table 2). The landmarks, curves, and curve semi-landmarks were placed either manually on the PF surface mesh using EVAN Toolbox software (v.1.71; www.evan-society.org) or semi-automatically using a dedicated custom-made software developed in MATLAB R2013a (Landmark 10, curves 3 and 4; Table 2; the software code is available via https://github.com/nirsh1/Bone-Analysis.git). Semi-landmark sliding along the curves was carried out to minimize the thin-plate spline (TPS) bending energy between the target and template[90].

**Statistics and reproducibility**. Reliability analyses were carried out by examining the intra- and inter-observer variations in the shape of femoral landmark configurations; they were assessed using four randomly selected femora. To assess the intra-observer variation, one researcher (H.L.A) placed the landmarks and curve semi-landmarks three times on each of the femora with a week-long interval between landmarking sessions. To assess the inter-observer variation, the set of landmarks was placed by an additional independent researcher (C.A). To examine variations in shape, principal components analysis (PCA) was carried out following a General Procrustes Analysis (GPA) that superimposes the coordinates of the landmarks and semi-landmarks, thus eliminating any differences in orientation, location, and size[91]. The significance of Procrustes distances within and between repeated measurements of specimens and by researchers was assessed via permutation tests (1,000 random permutations)[92].

For the 3D shape analysis, Cartesian coordinates were converted into shape variables through GPA. PCA or bgPCA was carried out to examine the shape variation in the studied populations. The average Procrustes distances were calculated for each group from the mean shape of each group and presented in a heatmap and dendrogram in R (using heatmap.2 library). A Procrustes ANOVA was carried out in R (using the geomorph library) to determine whether group shape differences were a manifestation of shape allometry (shape~ group * logarithm of the centroid size). Pairwise analyses were carried out to examine significant differences in shape variation between every two groups. Outliers were removed from the analysis following Cardini et al[93].

To test the classification of human PF to groups (prehistoric, recent humans, and ICHF) according to its shape, a linear discriminant function (LDA) using the Jack-knife method was carried out on the PC scores, which explained at least 70% of the shape variance. These PC scores were obtained from a PCA on the shape-space of the PF of the prehistoric, recent humans, and ICHF groups.

**Reporting summary**. Further information on research design is available in the Nature Portfolio Reporting Summary linked to this article.

## Data availability

Landmark and semi-landmark data following GPA for each analysis are available via the link https://figshare.com/s/be2e703abee51789216f. We have no authority to share CT

scans of the original fossils with a third party. Access to scans of the ancient human sample should be requested from H.M. Any other relevant data are available upon reasonable request.

## Code availability

The code used in this study is available via https://github.com/nirsh1/Bone-Analysis.git.

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

## Acknowledgements

This study is part of H.L.A's M.Sc. thesis from the Sackler Faculty of Medicine, Tel Aviv University, which was carried out under the supervision of Dr. Hila May. We would like to thank Mirriam Tawane, the Ditsong National Museum of Natural History for the South African specimens from Swartkrans as well as Matt Skinner. We would also like to thank the Ethiopian Authority for Research and Conservation of the Cultural Heritage (ARCCH) and its curatorial staff, as well as Zeresenay Alemseged, William Kimbel, Fred Spoor, Heiko Temming, and David Plotzki. We Thank Prof. Nathan Peled, the head of the Radiology Department, Elisha Hospital, Israel for providing access to CT scans of recent humans scanned at Carmel Medical Center, Israel. We would like to thank the Mäxi Foundation, Switzerland, the Broad Institute-Israel Science Foundation Program for Collaborative Projects (#2632/18), and the United States-Israel Binational Science Foundation (#2019041).

## Author contributions

H.L.L. collected the data, co-developed the methods, and co-wrote the manuscript; N.S. programmed the semi-automatic software for curve placement on the proximal femur and revised the manuscript; A.P. created the figures and revised the manuscript; S.F. and R.P.-K. edited and revised the manuscript; V.R. collected data and revised the manuscript; J.-J.H. contributed fossils to the study and revised the manuscript; F.R. co-designed the research and revised the manuscript; H.M. co-designed the research, co-developed the methods, carried out the statistical analyses, and wrote the manuscript.

## Competing interests

The authors declare no competing interests.
