## [Peer Review file · Communications Biology]

Evolutionary roots of the risk of hip fracture in humansReviewers' comments:

Reviewer #1 (Remarks to the Author):

The article is both well structured and well written, which reflects the scientific maturity of the authors. It seeks to approach some of the consequences of bipedality.

It is pretended to find arguments to prove that bone morphology is a fundamental risk factor for fracturing the hip. For that, the authors gathered the possible sample where some shortcomings were noted: whereas diverse specimens were included, from Pan troglodytes, Australopithecus, Paranthropus, and early Homo, among others, it is evident that there is a shortcoming with the sample size of those just mentioned specimens. Yet, I am aware that enlarging the sample is problematic.

Nevertheless, there are many more individuals nowadays, which makes the sample asymmetry notorious. This asymmetry restrains/limits the inferences about how the PF alterations initiated with the transition to bipedal locomotion continued in modern humans. The authors need to remember that there were several tries/attempts at bipedality, not only one. Furthermore, while some were dead ends and didn't make it, others, and not only one, were well accomplished. The way the arguments are presented in this article makes the readers think that there is only one pathway to bipedality. Besides, the first bipedal had incipient bipedality and kept another type of locomotion simultaneously. That said, I found statements such as « femoral size differences did not dictate femoral shape during human evolution » an extrapolation hard to justify because of the low nr of first bipedals in the sample.

I also found the conclusions not unexpected, not new. If there were several pathways to bipedality, with individuals living in different environments, there were obviously different ways of walking, a different combination of modes of locomotion, other balances among the two lower limbs, and different requirements.

If I found that the sample size of early bipedals a limiting factor, on the other hand, I also found the sample size for modern humans a very good one, allowing good conclusions like that sedentism led to shortening and widening AP of the femoral neck; or that the shortening of femoral neck increased the risk of hip fracture.

In all, while I do not consider it a novelty that ICHF is linked with more efficient bipedal walking and that reduced physical activity and sedentary lifestyles are important etiological factors, it is worthwhile to find it so well proven as it is in this article. That said, it should be considered for publication.

Reviewer #2 (Remarks to the Author):

Communications Biology Review

I have reviewed the manuscript 'Evolutionary roots of the risk of hip fracture in humans' by Avni et al. The manuscript addresses hip fracture risk within an evolutionary context, using palaeoanthropological and archaeological samples. The authors make good use of figure illustrations to assist in their argument and lead to a logical conclusion. There are a few comments that I believe need to be addressed, which I list below.

The first sentence of the abstract was a bit difficult to follow along smoothly. It would be a good idea to rewrite this sentence.

Line 41 – it would be better to word it as "Using Geometric Morphometric methods..."

It would be best to avoid abbreviations in the abstract.

The authors should be consistent when using scientific species names versus the common names. E.g., Homo sapiens versus Neanderthals. This happens throughout the manuscript and I urge the

authors to be consistent in terminology throughout (i.e., see abstract and line 122 as examples).

Lines 73-77: there's a mix of tenses within this sentence.

Line 78: It seems a bit out of place to mention that differences continued during the Holocene without previous mention to other time periods. Furthermore, if the target audience is medical researchers (including palaeoanthropologists), then I would recommend to also use rough dates when mentioning time periods.

The last sentence of the first paragraph seems to have been appended out of place – the paragraph doesn't quite flow coherently. I would recommend the authors to rewrite this first paragraph.

Line 82: technically speaking, humans and other hominins are also great apes.

Line 112: should avoid use of adverbs such as 'excellent'. Same line, perhaps wording such as 'intra- and inter-observer error was negligible...' should be used instead to follow wording choice of other geometric morphometric papers.

Line 116: I would urge the authors to find another way to discuss modern human differences. The language choice of "recent males and females" sounds out of place and also unclear as to whether they mean living populations or general *Homo sapiens* specimens. Leading on from that – would you expect to see sexual differences in other extant species? And how sexually dimorphic are the hip joints of hominin specimens? Some sort of discussion by the authors would be beneficial to ascertain if we should or shouldn't be concerned with potential sexual dimorphism. Also, what was the justification for limiting the age range to 18 to 45? Some studies have shown that the hip isn't fully developed until age 20. For the upper limit – is this the cut-off in which we would expect to see hip fracture prevalence increase, or rather the limits of their sample? More discussion and justification is needed.

Line 158: not clear what is meant by 'herders'? This isn't typical terminology used in palaeoanthropology. Perhaps hunter-gatherers would be better? If this specific group are typically referred to as herders, then the authors need to be clear.

Line 165: 9.8% of shape variation is so low that I'm not sure this is worth mentioning.

Line 242: The first sentence of the discussion doesn't make sense – is there a word missing after 'PF'?

Greater reference to previous literature in the discussion would be beneficial. For example, the following articles are key, but are not currently cited (to name a few):

Berge, C. (1994). How did the australopithecines walk? A biomechanical study of the hip and thigh of *Australopithecus afarensis*. *Journal of Human Evolution*, 26, 259-273.

McHenry, H. M. (1975). The ischium and hip extensor mechanism in human evolution. *American Journal of Physical Anthropology*, 43, 39-46.

Vidal-Cordasco, M., Mateos, A., Zorrilla-Revilla, G., Prado-Novoa, O., & Rodriguez, J. (2017). Energetic cost of walking in fossil hominins. *Am J Phys Anthropol*, 164(3), 609-622.

Wiseman, A. L. A., Demuth, O. E., Pomeroy, E., & De Groote, I. E. (2022). Reconstructing articular cartilage in the *Australopithecus afarensis* hip joint and the need for modelling six degrees of freedom. *Integrative Organismal Biology*, obac031.

Line 268: There's been greater debate as to obstetrics in human evolution. Recent literature must also be considered. See:

Frémondrière, P., Thollon, L., Marchal, F. et al. Dynamic finite-element simulations reveal early origin of complex human birth pattern. *Commun Biol* 5, 377 (2022).

Nathan E. Thompson, Danielle Rubinstein, William Parrella-O'Donnell, Matthew A. Brett, Brigitte Demes, Susan G. Larson, Matthew C. O'Neill; The loss of the 'pelvic step' in human evolution. *J Exp Biol* 15 August 2021; 224 (16): jeb240440.

Line 274: It would be beneficial for the authors to elaborate in text about bone loading and the impact this has on bone morphology.

Line 276: word repetition

Line 277: So far, the authors have not given adequate backing to their claim their 3D shape analysis provides improved sets of results over other methods. Whilst I am not dismissing their claim, this argument needs to be bolstered and contextualised with previous studies using other methods, alongside a discussion of potential study limitations.

Line 281-284: I would recommend that the authors revise the wording of this sentence, it is difficult to follow.

Line 285: the authors need to describe the concept of a moment arm alongside relevant citations. For example:

Pandy, M. G. (1999). Moment arm of a muscle force. *Exercise and Sport Sciences Reviews*, 27(1), 79-118.

van Beesel, J., Hutchinson, J. R., Hublin, J. J., & Melillo, S. M. (2021). Exploring the functional morphology of the Gorilla shoulder through musculoskeletal modelling. *J Anat*, 239(1), 207-227.

Furthermore, I would also like to ask the authors what the limitations of their study might be as no soft tissues have been modelled. If the authors were instead to use dynamic simulations to model mobility alongside finite element analysis of joint surface contact, would the results hold up? (This could be a future step of their study and in no way am I recommending that this should be done here). For example, see: Xiong, B., Yang, P., Lin, T. et al. Changes in hip joint contact stress during a gait cycle based on the individualized modeling method of "gait-musculoskeletal system-finite element". *J Orthop Surg Res* 17, 267 (2022).

Line 288: do the authors mean that the gluteals mainly resist abduction? They are hip abductors, so are antagonistic to adduction.

Line 289: moment and moment arm used interchangeably and this is incorrect. A muscle's moment arm is defined as the perpendicular distance between the muscle's line of action and the joint axis, representing the effectiveness with which the force produced by a muscle generates torques at the joint(s) crossed by the muscle. Only a generalisation of muscular function and capability can be ascertained from moment arms (although I am struggling to see the relevance here as this study has not modelled any soft tissues, nor digitised muscle attachment sites). Moment arms are intrinsically linked to bone shape and size, and also the path of the muscle within the body (all three of these need to be considered; see Pandy 1999), but currently the description in the discussion section has not quite 'hit the mark'. Please reword for clarity and accuracy. Further, the statement that the lever arm ratio of the abductors is required for gait is somewhat wrong. Yes, the abductors are pivotal in locomotion, but they are only three muscles acting to move the hip. Many other muscles are equally important for gait, not just the abductors. The internal rotator compartment prevents rotation of the body (and thus body destabilisation), whilst the extensors and flexors are – arguably – the muscles facilitating the actual capability of forward gait – overall, the word choice by the authors greatly oversimplifies muscular function. Relating this to fracture risk of the proximal femur, surely the internal/external compartments are equally as important as the abductors because they are the muscles counter-acting body tilt and thus failure of these muscles can lead to undue stress on the

femoral neck? Overall, my opinion is to remove this section because it doesn't fit with the rest of the discussion.

Line 295-302: what's the risk fracture of a chimpanzee which has a really long neck? That would help strengthen the argument in this paragraph.

Line 307 – Homo not italicised.

Line 350: please define what is meant by ancient. I think prehistoric would be better.

Line 376: DEXA will need to be discussed in more detail. Also, because you are obtaining information from their medical files, details regarding ethical approval and conduct will need to be supplied.

Line 401: is the matlab code custom made? Or have you used another package? If the former, will it be published alongside the paper?

The variation in the PCA of hominini prox femur shape (Fig 1a) is quite large! Can the authors please provide a figure alongside the table so that the reader may visualise landmark placement.

Whilst I really like figure 2c (very pretty figures!), I am struggling to see the relevance of this figure within the grand scheme of the paper.

Whilst there's ~no difference along PC2 in Fig3a, there is quite a difference between the early hominins and early Homo along PC1, but the PC score is quite low (26.8%). Can the authors double check these numbers?

For Figure 4, the authors may wish to flatten their image. Currently, there are white box lines cutting through the femurs (I assume these are not meant to be there).

Reviewer #3 (Remarks to the Author):

This is an interesting and well-written paper worthy of publication. I like the fact that the authors are tackling this evolutionary issue, and are tying it to current medical problems. Their approach is creative and worthwhile. There are, however, some statements made that I would ask them to reconsider or justify more thoroughly than they have so far.

My biggest critique is RE: page 7, lines 149-151 "It is noteworthy that although slight allometry existed within each group, the groups did not share a common allometric trajectory (Supplementary Fig. S3), meaning that femoral size differences did not dictate femoral shape during human evolution." How can the authors make this assumption? Are they assuming that *P. troglodytes* represents the ancestral condition? We have been burned so many times on this assumption that surely by now we've learned our lesson. Chimpanzees are highly derived in their locomotor anatomy, and if anything, hominins may be closer in morphology to the LCA than any of the African ape species. Second, I would not recommend reconstructing allometric trajectories based on three fossil Homo specimens and four Australopithecus/Paranthropus ones. Additionally, we cannot be certain any of the latter group are ancestral to *H. sapiens* (and we know that Paranthropus is not!).

I think the authors are on solid ground when they discuss the morphology that is more likely to be associated with hip fractures, and the fact that these femora, while within the recent human range, nonetheless tend to be at its margins. As such, I do not see why the allometric argument they make is

even relevant. I would recommend excising it altogether.

My second critique is perhaps more a question than a critique, per se. I learned years ago that in many osteoporotic hip fracture cases, the hip fractures first, causing the patient to fall. The patient will frequently say "I fell and broke my hip," but what has really occurred is that the patient's hip broke, causing her to fall. I am not a clinician; is what I learned wrong or woefully out of date? Are most hip fractures now known to be the direct result of a lateral fall?

Minor corrections:

Abstract: I think "This study aims" works better. Similarly, I would suggest saying We "show" - I don't know why the past tense is being used in the abstract. The reader is reading it now - it should be in the present tense.

p. 11, first two paragraphs of the Discussion, "Homo" is not italicized 3 times, and in line 259, should be reworded to read "genus Homo" instead of "Homo genus" - it just sounds better.

Revised Manuscript COMMSBIO-22-2749-T

We thank the reviewers for their constructive comments, which were highly insightful. We believe that their comments improved our paper.

Response to reviewers

Reviewer #1 (Remarks to the Author):

1. The article is both well structured and well written, which reflects the scientific maturity of the authors.

Answer: We thank the reviewer for this comment.

2. It seeks to approach some of the consequences of bipedality. It is pretended to find arguments to prove that bone morphology is a fundamental risk factor for fracturing the hip. For that, the authors gathered the possible sample where some shortcomings were noted: whereas diverse specimens were included, from Pan troglodytes, Australopithecus, Paranthropus, and early Homo, among others, it is evident that there is a shortcoming with the sample size of those just mentioned specimens. Yet, I am aware that enlarging the sample is problematic. Nevertheless, there are many more individuals nowadays, which makes the sample asymmetry notorious. This asymmetry restrings/limits the inferences about how the PF alterations initiated with the transition to bipedal locomotion continued in modern humans.

Answer: As the reviewer commented, the sample size is a familiar limitation in paleoanthropology studies. In this study, allocating the proximal femora of fossils that would fulfill our inclusion criteria was challenging, and we included all femora that we are aware of that corresponded to our inclusion criteria (detailed in the materials section in the revised manuscript). The reviewer is correct in commenting that the biased sample size impedes our ability to describe the adaptation to bipedality throughout human evolution as a linear process. Indeed, aware of these limitations, in the revised manuscript, we refrained from presenting differences in the proximal femoral morphology between groups as a linear process. We rephrased the description of our abstract, results, and discussion accordingly (e.g., p. 5, lines: 69-70; p. 8, lines: 322-325). In addition, we added a study limitations section to the revised manuscript relating to this issue (p. 20, lines 424-430).

3. The authors need to remember that there were several tries/attempts at bipedality, not only one. Furthermore, while some were dead ends and didn't make it, others, and not only one, were well accomplished. The way the arguments are presented in this article makes the readers think that there is only one pathway to bipedality. Besides, the first bipedal had incipient bipedality and kept another type of locomotion simultaneously.

Answer: The reviewer is right; the transition to bipedal locomotion was a complex process. We rephrased the text to comply with this comment (p.5, lines: 73-74; p. 16-17, lines: 330-346).

4. That said, I found statements such as « femoral size differences did not dictate femoral shape during human evolution » an extrapolation hard to justify because of the low nr of first bipedals in the sample.

Answer: The first and third reviewers were in agreement regarding this issue, and we agree with their comments. Therefore, we adopted the third reviewer's recommendation and rephrased this section. In the revised manuscript, we only referred to the allometric effect between humans and *pan troglodytes*.

5. I also found the conclusions not unexpected, not new. If there were several pathways to bipedality, with individuals living in different environments, there were obviously different ways of walking, a different combination of modes of locomotion, other balances among the two lower limbs, and different requirements.

Answer: Vast research has been carried out regarding the morphological adaptation of the PF to bipedalism during human evolution, also emphasizing different walking modalities (as cited in the text). Since we applied a new GM protocol in this research, we wanted to examine the correspondence between the results obtained using this protocol and previous ones. We would have been surprised if we had found completely different morphological differences using our protocol. The compatibility we found between our results and previous ones, especially those using the landmark-based GM method, strengthened our perception that we are on safe ground to test new hypotheses, which were the major aims of our study. Namely, were there changes in the PF shape between human groups with different subsistence strategies? Does PF

shape play a role in the risk of intracapsular hip fracture? This point is emphasized in the revised manuscript (p. 17, lines 360-361).

6. If I found that the sample size of early bipedals a limiting factor, on the other hand, I also found the sample size for modern humans a very good one, allowing good conclusions like that sedentism led to shortening and widening AP of the femoral neck; or that the shortening of femoral neck increased the risk of hip fracture.

Answer: We thank the reviewer for this comment.

7. In all, while I do not consider it a novelty that ICHF is linked with more efficient bipedal walking and that reduced physical activity and sedentary lifestyles are important etiological factors, it is worthwhile to find it so well proven as it is in this article. That said, it should be considered for publication.

Answer: We thank the reviewer for recommending to consider the manuscript for publication.

Reviewer #2 (Remarks to the Author):

Communications Biology Review

1. I have reviewed the manuscript 'Evolutionary roots of the risk of hip fracture in humans' by Avni et al. The manuscript addresses hip fracture risk within an evolutionary context, using palaeoanthropological and archaeological samples. The authors make good use of figure illustrations to assist in their argument and lead to a logical conclusion. There are a few comments that I believe need to be addressed, which I list below.

Answer: We thank the reviewer for this comment.

2. The first sentence of the abstract was a bit difficult to follow along smoothly. It would be a good idea to rewrite this sentence.

Answer: Following the reviewer's comment, we rewrote the first sentence.

3. Line 41 – it would be better to word it as "Using Geometric Morphometric methods..."

Answer: We corrected it as suggested by the reviewer.

4. It would be best to avoid abbreviations in the abstract.

Answer: As requested by the reviewer, we reduced the usage of abbreviations in the abstract.

5. The authors should be consistent when using scientific species names versus the common names. E.g., *Homo sapiens* versus Neanderthals. This happens throughout the manuscript and I urge the authors to be consistent in terminology throughout (i.e., see abstract and line 122 as examples).

Answer: Following the reviewer's suggestion, we changed the term *Homo sapiens* to humans throughout the text of the revised manuscript.

6. Lines 73-77: there's a mix of tenses within this sentence.

Answer: We thank the reviewer for bringing this to our attention; we corrected this error. In addition, a professional English editor revised the manuscript.

7. Line 78: It seems a bit out of place to mention that differences continued during the Holocene without previous mention to other time periods. Furthermore, if the target audience is medical researchers (including palaeoanthropologists), then I would recommend to also use rough dates when mentioning time periods.

Answer: Following the reviewer's comment, we rephrased this sentence. We also provided the timing of the period mentioned (p. 5, lines 79-80).

8. The last sentence of the first paragraph seems to have been appended out of place – the paragraph doesn't quite flow coherently. I would recommend the authors to rewrite this first paragraph.

Answer: Following the reviewer's suggestion, we rewrote the first paragraph. We believe that in the revised manuscript the text flows.

9. Line 82: technically speaking, humans and other hominins are also great apes.

Answer: We rephrased this sentence and related to them as hominids as it is the term given to all modern and extinct great apes, including humans, chimpanzees, gorillas, orangutans, and their ancestors (p. 5, line: 78).

10. Line 112: should avoid use of adverbs such as 'excellent'. Same line, perhaps wording such as 'intra- and inter-observer error was negligible...' should be used instead to follow wording choice of other geometric morphometric papers.

Answer: We rephrased the sentence, as suggested by the reviewer.

11. Line 116: I would urge the authors to find another way to discuss modern human

differences. The language choice of "recent males and females" sounds out of place and also unclear as to whether they mean living populations or general Homo sapiens specimens.

Answer: As suggested by the reviewer, we rephrased this sentence in the revised manuscript (p. 6, line 116).

12. Leading on from that – would you expect to see sexual differences in other extant species? And how sexually dimorphic are the hip joints of hominin specimens? Some sort of discussion by the authors would be beneficial to ascertain if we should or shouldn't be concerned with potential sexual dimorphism.

Answer: Unfortunately, although there are several studies on the sexual dimorphism of extinct hominins based on size measures, there are no data regarding the sexual dimorphism of proximal femoral shape. This is also true for chimpanzees. In the revised manuscript, we added analysis of shape variance among the chimpanzees by sex, as well as statistical analyses (Procrustes ANOVA and pairwise comparison; Supplementary Fig. S2). Male and female chimpanzees, similar to humans, did not differ significantly in their PF shape variance (p. 6-7, lines: 117-121). Although we could not directly examine sexual dimorphism among extinct hominins due to the small sample size and uncertainty regarding their sex, we feel confident that they followed a similar trend. In the revised manuscript, we provided additional data to support our decision to carry out the analysis when sex is combined. Furthermore, we discussed this issue in the first paragraph of the discussion (p. 15-16, lines; 303-313).

13. Also, what was the justification for limiting the age range to 18 to 45? Some studies have shown that the hip isn't fully developed until age 20. For the upper limit – is this the cut-off in which we would expect to see hip fracture prevalence increase, or rather the limits of their sample? More discussion and justification is needed.

Answer: The reviewer is right; in some individuals, only at the age of 21 will the femoral head completely fuse with the femoral neck, yet 95% of individuals over the age of 18 will have a fully developed femur (Schaefer et al., 2009). In our study, only individuals with a completely fused femoral head were included in the study in both the prehistoric and recent samples (we

added this criterion in the exclusion criteria of the study; p. 21, line 474-475). Furthermore, the PF shape variance of the 18-21 and 22-45 age groups overlap considerably (see the figure below), with no significant difference between the group.

Depicting the upper age limit was indeed more complicated. However, two main justifications guided us:

- 1) Differences in life expectancy between the ancient and recent groups: Although we could not estimate the exact or even a narrow age range for most ancient individuals, according to the literature, life expectancy in prehistory (~30 years; Supplementary Table S1) was much lower than that of the Israeli population from which our recent sample derived. Although age had little effect on the PF shape (Table 1), we preferred to examine, as much as possible, groups with similar demographic characteristics. The age of 45 was chosen because it is still considered an age with a low prevalence of diseases such as osteoporosis and osteoarthritis (Hernandez et al., 2003; Cross et al., 2014).
- 2) Unequal sample size: The fossil, ancient, and chimpanzee samples were much smaller than the recent sample. The strength of this study lies in the large sample size of the recent population. This issue was also mentioned by the first reviewer, who commented: "I also found the sample size for modern humans a very good one, allowing good conclusions...". However, to reduce bias due to unequal sample sizes, we compared groups with a common age cohort base.

Following the reviewer's request, we added justifications for subsampling the recent population in the supplementary information (Supplementary Table S2).

Figure 1: Principal Component Analysis of shape space of recent individuals aged 18-21 (blue) and 22-45 years (green).

14. Line 158: not clear what is meant by 'herders'? This isn't typical terminology used in palaeoanthropology. Perhaps hunter-gatherers would be better? If this specific group are typically referred to as herders, then the authors need to be clear.

Answer: Following the reviewer's request, we changed the Chalcolithic Protohistoric sample label to pastoralists throughout the revised manuscript. We believe that this term will be clearer to the reader.

15. Line 165: 9.8% of shape variation is so low that I'm not sure this is worth mentioning.

Answer: The reviewer is right that 9.8% is low; however, we prefer to leave this number because it demonstrated that group membership has a greater effect (almost 3 times more) on shape variance than does the PF size.

16. Line 242: The first sentence of the discussion doesn't make sense – is there a word missing after 'PF'?

Answer: We rephrased this sentence.

17. Greater reference to previous literature in the discussion would be beneficial. For example, the following articles are key, but are not currently cited (to name a few):

Berge, C. (1994). How did the australopithecines walk? A biomechanical study of the hip and thigh of *Australopithecus afarensis*. *Journal of Human Evolution*, 26, 259-273.

McHenry, H. M. (1975). The ischium and hip extensor mechanism in human evolution. *American Journal of Physical Anthropology*, 43, 39-46.

Vidal-Cordasco, M., Mateos, A., Zorrilla-Revilla, G., Prado-Novoa, O., & Rodriguez, J. (2017). Energetic cost of walking in fossil hominins. *Am J Phys Anthropol*, 164(3), 609-622.

Wiseman, A. L. A., Demuth, O. E., Pomeroy, E., & De Groote, I. E. (2022). Reconstructing articular cartilage in the *Australopithecus afarensis* hip joint and the need for modelling six degrees of freedom. *Integrative Organismal Biology*, obac031.

Answer: The references suggested by the reviewer were included in the introduction and discussion.

18. Line 268: There's been greater debate as to obstetrics in human evolution. Recent literature must also be considered. See:

Frémondrière, P., Thollon, L., Marchal, F. et al. Dynamic finite-element simulations reveal early origin of complex human birth pattern. *Commun Biol* 5, 377 (2022).

Nathan E. Thompson, Danielle Rubinstein, William Parrella-O'Donnell, Matthew A. Brett, Brigitte Demes, Susan G. Larson, Matthew C. O'Neill; The loss of the 'pelvic step' in human evolution. *J Exp Biol* 15 August 2021; 224 (16): jeb240440.

Answer: Following the reviewer's comment, we added the references suggested and elaborated on the association between the femoral morphological changes and bipedality and the impact of changes in pelvis morphology following the transition to bipedality regarding obstetrics.

19. Line 274: It would be beneficial for the authors to elaborate in text about bone loading and the impact this has on bone morphology.

Answer: We elaborated on how bone loading impacted its morphology as suggested by the reviewer (p. 17; lines 349-351).

20. Line 276: word repetition

Answer: The manuscript underwent an additional revision by a professional English editor.

21. Line 277: So far, the authors have not given adequate backing to their claim their 3D shape analysis provides improved sets of results over other methods. Whilst I am not dismissing their claim, this argument needs to be bolstered and contextualised with previous studies using other methods, alongside a discussion of potential study limitations.

Answer: Most studies examined differences in the proximal femur between hominin groups based on linear measurements. To the best of our knowledge, only Harmon (2009) used both methods (i.e., linear and landmark-based geometric morphometrics) on the same sample; however, no direct comparison between the methods was carried out. Nevertheless, it was mentioned that new morphological data could be acquired via the landmark-based geometric morphometric method. Furthermore, the results in that paper yielded better differentiation between groups using the geometric morphometric method. To reduce confusion, we rephrased this sentence and lowered the tone (p. 17, lines 360-361). We added a study limitation section as requested (p. 20).

22. Line 281-284: I would recommend that the authors revise the wording of this sentence, it is difficult to follow.

Answer: This sentence was rephrased as suggested by the reviewer.

23. Line 285: the authors need to describe the concept of a moment arm alongside relevant citations. For example:

Pandy, M. G. (1999). Moment arm of a muscle force. *Exercise and Sport Sciences Reviews*, 27(1), 79-118.

van Beesel, J., Hutchinson, J. R., Hublin, J. J., & Melillo, S. M. (2021). Exploring the functional morphology of the Gorilla shoulder through musculoskeletal modelling. *J Anat*, 239(1), 207-227.

25. Line 288: do the authors mean that the gluteals mainly resist abduction? They are hip abductors, so are antagonistic to adduction.

26. Line 289: moment and moment arm used interchangeably and this is incorrect. A muscle's moment arm is defined as the perpendicular distance between the muscle's line of action and the joint axis, representing the effectiveness with which the force produced by a muscle generates torques at the joint(s) crossed by the muscle. Only a generalisation of muscular function and capability can be ascertained from moment arms (although I am struggling to see the relevance here as this study has not modelled any soft tissues, nor digitised muscle attachment sites). Moment arms are intrinsically linked to bone shape and size, and also the path of the muscle within the body (all three of these need to be considered; see Pandy 1999), but currently the description in the discussion section has not quite 'hit the mark'. Please reword for clarity and accuracy. Further, the statement that the lever arm ratio of the abductors is required for gait is somewhat wrong. Yes, the abductors are pivotal in locomotion, but they are only three muscles acting to move the hip. Many other muscles are equally important for gait, not just the abductors. The internal rotator compartment prevents rotation of the body (and thus body destabilisation), whilst the extensors and flexors are – arguably – the muscles facilitating the actual capability of forward

gait – overall, the word choice by the authors greatly over-simplifies muscular function. Relating this to fracture risk of the proximal femur, surely the internal/external compartments are equally as important as the abductors because they are the muscles counter-acting body tilt and thus failure of these muscles can lead to undue stress on the femoral neck? Overall, my opinion is to remove this section because it doesn't fit with the rest of the discussion.

Answer: Following the reviewer's comments (23, 25, and 26), we reevaluated the essence of this paragraph. Indeed, we found it confusing and missing the point of the discussion, mainly because the data available from recent humans are vastly debated (as currently mentioned in the text). Therefore, a biomechanical explanation for a specific association (femoral length and risk of fracture) might be misleading and doesn't contribute to our discussion because we focused on other characteristics. Furthermore, expanding the discussion on this issue is out of the scope of our paper. Therefore, we omitted this paragraph from the revised manuscript as advised by the reviewer and rephrased the discussion.

24. Furthermore, I would also like to ask the authors what the limitations of their study might be as no soft tissues have been modelled. If the authors were instead to use dynamic simulations to model mobility alongside finite element analysis of joint surface contact, would the results hold up? (This could be a future step of their study and in no way am I recommending that this should be done here). For example, see: Xiong, B., Yang, P., Lin, T. et al. Changes in hip joint contact stress during a gait cycle based on the individualized modeling method of "gait-musculoskeletal system-finite element". J Orthop Surg Res 17, 267 (2022).

Answer: This is a very interesting question, and we thank the reviewer for raising it. Following the results of our study, additional research is now being conducted using finite element analysis on the mean shapes we have extracted from our study. However, as indicated by the references the reviewer recommended, a good question is whether it will hold in 'real' life. We will definitely think about how to explore this issue further. A section relating to the study limitations was added to the revised manuscript (p. 20).

27. Line 295-302: what's the risk fracture of a chimpanzee which has a really long neck? That would help strengthen the argument in this paragraph.

Answer: This is an excellent question. We tried to answer it when we started the study. However, hip fractures in chimpanzees are very rare (p. 18, line 373). Therefore, we did not have any chimpanzees with hip fractures to study. Calculating the hip fracture risk of a chimpanzee using a comparative sample of humans is problematic due to the large differences in PF shape between the species (which will yield a chance of 0). However, the reviewer's question encouraged us to test the chance of different human groups being categorized into the hip fracture group. Accordingly, none of the prehistoric individuals were classified in the hip fracture group, whereas 11% were classified as recent humans. Nevertheless, 15.5% of the recent humans without a fracture were classified in the hip fracture group. These results were added to the revised manuscript (results: p. 13, lines 263-265, discussion: p. 18, lines 367-368, and methods: p. 22, lines 515-519) and to the Supplementary Information (Supplementary Table S5) of the revised manuscript.

28. Line 307 – Homo not italicised.

Answer: We italicised *Homo*.

29. Line 350: please define what is meant by ancient. I think prehistoric would be better.

Answer: We followed the reviewer's suggestion.

30. Line 376: DEXA will need to be discussed in more detail. Also, because you are obtaining information from their medical files, details regarding ethical approval and conduct will need to be supplied.

Answer: We provided more details from where the DEXA scores were obtained as well as how the diagnosis was made. We also expanded the details of the ethical approval (p. 21, lines 461-464, 473-475).

31. Line 401: is the matlab code custom made? Or have you used another package? If the former, will it be published alongside the paper?

Answer: We clarified in the text that it is custom-made software (written by Prof. Shvalb). The code will be published; along with the paper we provided a link with access to the code (p. 22, lines 490-491).

32. The variation in the PCA of hominini prox femur shape (Fig 1a) is quite large! Can the authors please provide a figure alongside the table so that the reader may visualise landmark placement.

Answer: We followed the reviewer's suggestion. In the revised manuscript, Figure 1 includes the PF model with landmarks, curves, and semilandmarks (Fig. 1c) as well as landmark definitions (Figure 1d). Table 2 includes the curves' position definition. Following this change, we omitted the figure of the PF with the landmarks and semilandmarks from the supplementary information.

The revised figure:

Fig. 1 | **a**, Principal Component Analysis (PCA) plot in shape-space for the proximal femur of *Pan troglodytes* (orange), *Australopithecus/Paranthropus* (light blue), early *Homo* (light green), Neanderthals (red), and recent humans (gray; age 18-45 years). PC1 (explains 24.4% of the shape variance) distinguishes between *Pan troglodytes* and recent humans; early hominins and early *Homo* fell in between and can be distinguished from other groups along PC2 (explains 21.6% of the shape variance). Neanderthals are in the upper variation of recent humans along both PC1 and PC2. **b**, PCA in shape-space for the proximal femur among populations with different subsistence strategies: Hunter-gatherers – Epi-Paleolithic (purple), early farmers – Pre-Pottery Neolithic (blue), pastoralists – Chalcolithic (pink), and recent humans (gray; 18-45 years old). A gradient of change in the proximal femoral shape variance over time is evident along PC1. Details regarding the sample included in the study appear in Supplementary Tables S1 and S2. **c**, The positions of landmarks (orange, numbered dots), curves (green lines), and semilandmarks (blue

dots) on the proximal femora. **d**, Definition of the landmarks' position (definitions of the curves' position are presented in Table 2).

33. Whilst I really like figure 2c (very pretty figures!), I am struggling to see the relevance of this figure within the grand scheme of the paper.

Answer: This figure demonstrates the magnitude of differences between the mean shape of each group from the proximal femur obtained from the non-osteoporotic, non-fractured group. In addition, it demonstrates the location of the changes. We believe the value of this image is that it facilitates understanding the features related to an increased risk for intracapsular hip fracture and emphasizes that the differences in PF shape in the osteoporotic group are not related to those that increase the risk for hip fracture. We highlighted this point in the results section, and we also referred to it in the discussion (p. 19, lines 416-419).

34. Whilst there's ~no difference along PC2 in Fig3a, there is quite a difference between the early hominins and early Homo along PC1, but the PC score is quite low (26.8%). Can the authors double check these numbers?

Answer: We re-checked the numbers. PC1 indeed explains 26.8% of the variation. This number corresponds to other results presented in this study.

35. For Figure 4, the authors may wish to flatten their image. Currently, there are white box lines cutting through the femurs (I assume these are not meant to be there).

Answer: We thank the reviewer for pointing out this problem. We corrected the figure.

Reviewer #3 (Remarks to the Author):

This is an interesting and well-written paper worthy of publication. I like the fact that the authors are tackling this evolutionary issue, and are tying it to current medical problems. Their approach is creative and worthwhile. There are, however, some statements made that I would ask them to reconsider or justify more thoroughly than they have so far.

Answer: We thank the reviewer for this comment.

1. My biggest critique is RE: page 7, lines 149-151 "It is noteworthy that although slight allometry existed within each group, the groups did not share a common allometric trajectory (Supplementary Fig. S3), meaning that femoral size differences did not dictate femoral shape during human evolution." How can the authors make this assumption? Are they assuming that *P. troglodytes* represents the ancestral condition? We have been burned so many times on this assumption that surely by now we've learned our lesson. Chimpanzees are highly derived in their locomotor anatomy, and if anything, hominins may be closer in morphology to the LCA than any of the African ape species. Second, I would not recommend reconstructing allometric trajectories based on three fossil *Homo* specimens and four *Australopithecus/Paranthropus* ones. Additionally, we cannot be certain any of the latter group are ancestral to *H. sapiens* (and we know that *Paranthropus* is not!).

I think the authors are on solid ground when they discuss the morphology that is more likely to be associated with hip fractures, and the fact that these femora, while within the recent human range, nonetheless tend to be at its margins. As such, I do not see why the allometric argument they make is even relevant. I would recommend excising it altogether.

Answer: The reviewer is right; *Pan troglodytes* are not our ancestors, and we did not try to imply that. This is also why we did not examine the risk for hip fracture among chimpanzees. The reviewer is also right in claiming that reconstructing allometric trajectories, based on three fossils of the genus *Homo* and four *Australopithecus/Paranthropus* ones, is not valid. However, since allometry can also be an explanatory factor, we preferred to refer to it in the paper.

Following the reviewer's comment, we considered size as an explanatory factor for differences in proximal femoral shape only between humans and *Pan troglodytes*. We believe that it is important to mention it so readers won't wonder about the allometric effect. Therefore, we rephrased this section, eliminating the paragraph relating to allometry and human evolution as suggested. We kept the main results of the Procrustes ANOVA (the end of the first paragraph of this section) and indicated that differences in PF size between *Pan troglodytes* and humans did not dictate the differences in PF shape between them (the second paragraph).

2. My second critique is perhaps more a question than a critique, per se. I learned years ago that in many osteoporotic hip fracture cases, the hip fractures first, causing the patient to fall. The patient will frequently say "I fell and broke my hip," but what has really occurred is that the patient's hip broke, causing her to fall. I am not a clinician; is what I learned wrong or woefully out of date? Are most hip fractures now known to be the direct result of a lateral fall?

Answer: According to the literature, as well as from close working relationships with orthopedics specializing in hip replacement surgeries, the main known cause is a sideways fall, as we cited in our manuscript. Having said that, we agree with the reviewer's suggestion, although there is little literature to back us up (e.g., Cotton et al., 1994), and the literature available is based on models, not on epidemiologic studies. However, our results support this hypothesis because we demonstrated a different morphology between the non-osteoporotic hip fracture group and the non-fractured osteoporotic group. In the osteoporotic group, the main changes in the proximal femur (compared to the control group) included thinning of the anteroposterior dimension of the proximal end of the femoral neck. This might indicate that following abrupt changes in load, or an inadequate movement that places a large load on the proximal femur, a fracture while standing could occur and thus cause falling (i.e., first fracturing, then falling). We refer to this issue in the revised discussion (p. 19; lines 415-419).

Minor corrections:

4. Abstract: I think "This study aims" works better. Similarly, I would suggest saying We "show" - I don't know why the past tense is being used in the abstract. The reader is reading it now - it should be in the present tense.

Answer: The revised manuscript was edited by a professional English editor.

5. p. 11, first two paragraphs of the Discussion, "Homo" is not italicized 3 times, and in line 259, should be reworded to read "genus Homo" instead of "Homo genus" - it just sounds better.

Answer: We corrected these mistakes, as advised by the reviewer.

REVIEWERS' COMMENTS:

Reviewer #1 (Remarks to the Author):

The authors did a very good job in the rebuttal letter, where they thoroughly justified their options and made the changes accordingly to the comments. From my side, it is accepted.

Reviewer #2 (Remarks to the Author):

I have re-reviewed this manuscript and can confirm that the authors have made substantial edits/revisions to their manuscript following the first round of revision. These changes have greatly improved the quality and readability of the manuscript, and the authors should be applauded for their efforts. I now believe that the manuscript can be accepted, following some very minor editorial suggestions which I detail below. Good job!

Editorial comments:

Please check with the editorial team on the journal preferences before any changes, but I do feel that adding the word "henceforth" before each of the abbreviations would make for easier reading.

Line 82 "10 thousand" would be best written as "10,000"

Line 127: would read better as "when comparing prehistoric and living groups"

Figure 2 – I'm not sure if this figure has been compressed for the pdf (if so, ignore this comment), but please check the quality of the text in the legend. Currently, it's quite pixelated and hard to read.

Line 306: the phrasing "scattered throughout human evolution" sounds out of place. Perhaps the time periods could be listed instead?

Line 313: rather than 'gathered together', would read better as 'grouped together'

Line 342: the phrase "due to gaining new characteristics" implies that it was almost an active choice on how they developed. I would recommend that this be re-phrased.

Line 343: should read "resulted from an increase in brain size"

Line 347: should read "in uncommitted bipeds"

Line 414: should this sentence have a citation?

Reviewer #3 (Remarks to the Author):

Thank you for taking my comments into account. I believe the article is ready for publication.

Trenton Holliday